# Untousling the Role of Tousled-like Kinase 1 in DNA Damage Repair

**DOI:** 10.3390/ijms241713369

**Published:** 2023-08-29

**Authors:** Ishita Ghosh, Arrigo De Benedetti

**Affiliations:** Department of Medicine, Department of Biochemistry, Louisiana Health Science Center-Shreveport, Shreveport, LA 71103, USA; ishita.ghosh@ucsf.edu

**Keywords:** DNA damage and repair, TLKs, cancer, cell cycle, therapy and resistance, mitochondria and apoptosis

## Abstract

DNA damage repair lies at the core of all cells’ survival strategy, including the survival strategy of cancerous cells. Therefore, targeting such repair mechanisms forms the major goal of cancer therapeutics. The mechanism of DNA repair has been tousled with the discovery of multiple kinases. Recent studies on tousled-like kinases have brought significant clarity on the effectors of these kinases which stand to regulate DSB repair. In addition to their well-established role in DDR and cell cycle checkpoint mediation after DNA damage or inhibitors of replication, evidence of their suspected involvement in the actual DSB repair process has more recently been strengthened by the important finding that TLK1 phosphorylates RAD54 and regulates some of its activities in HRR and localization in the cell. Earlier findings of its regulation of RAD9 during checkpoint deactivation, as well as defined steps during NHEJ end processing, were earlier hints of its broadly important involvement in DSB repair. All this has opened up new avenues to target cancer cells in combination therapy with genotoxins and TLK inhibitors.

## 1. Introduction

Tousled-like kinases (TLKs) are Ser/Thr kinases which were discovered to have potential functions in DNA replication and in the DNA damage repair nexus in higher eukaryotes [1,2,3,4]. There are two homologs, tousled-like kinases 1 and 2 (TLK1 and TLK2), which share 94% amino acid sequence identity in the kinase domain and an overall 84% identity. There is significant overlap between the substrates of TLK1 and TLK2 so far obtained from in vitro mass spectrometry in different studies [5]. However, discovery of distinct biological roles of the paralogs is forthcoming, as not all their substrates are common. Their function overall has been addressed by viability studies with mouse embryos, which showed that TLK1 is not essential during the placental development stages, whereas TLK2 is required for placental development. Rather, the biological role of TLK1 was implicated in sensitivity to DNA damage induction, more so in the conditional KO of both proteins. The most well-supported role of TLK1 is in DNA damage response and replication. TLK1 depletion has been shown to delay S-phase progression [6]. This is possible because TLK1 has been found to interact with human RAD9 during replication fork stalling. Phosphorylation of RAD9 (S328) by TLK1 has been shown to dissociate the 9–1–1 complex, thereby causing cytosolic localization of RAD9, and thus participates in the deactivation of the checkpoint after completion of DNA repair [7]. There has been increasing appreciation for structural domains in kinases that can act as scaffolding units to recruit substrates for their catalytic function. Interestingly, TLK1 also possesses chaperone functionality that recruits RAD9 at the junction of dsDNA and ssDNA at the double-strand break (DSB) site [8]. Another downstream target of TLK1 is Asf1a/b, which is phosphorylated at its C-terminus during the S phase and leads to binding of the H3–H4 tetramer that is assembled on newly replicated DNA. TLK1 can directly bind to chromatin, and its interaction with chromatin has also been linked to replication stress. During replication stress, TLK1 binding to chromatin decreases [8,9]. TLK1B, a splice variant of TLK1, is expressed upon ionizing radiation (IR) exposure and provides radio resistance to cells. TLK1B overexpression has also been shown to induce UV resistance [10]. TLK1 and TLK1B share the conserved kinase domain, and, therefore, TLK1 and TLK1B have considerable substrate overlap. One such substrate is Asf1b, which is phosphorylated by TLK1B and has been shown to gain chromatin remodeling activity in cell extracts [10]. Similarly, TLK1-dependent phosphorylation of Asf1a leads to an increase in histone octamer recruitment at newly repaired DNA strands [9]. In another model organism, *T. brucei*, TLK1 interacted with Asf1a and Asf1b to phosphorylate the histone chaperones, which helped to maintain their activity [6].

## 2. Role of TLK1 in DSB Repair

In vitro, tousled-like kinase 1 can bind directly with many DNA-damage- and DSB-repair-related proteins such as NEK1, AKTIP, RAD54B, FANCM [11], and even APEX2 with its established role in synthetic lethality with BRCA2 deficiency [12]. A more specific role of TLK1 in regulating DSB repair via RAD54 has recently been discovered [4]. This is further addressed below, but RAD54, along with its partner RAD51 (the key recombinase of eukaryotes), performs much of the work in the HRR process, which has led to RAD54 being named the Swiss army knife of HRR [13] (Figure 1).

Upon DSB induction by IR, TLK1 is transiently inhibited via a Chk1-dependent mechanism, and, hence, depletion of Chk1 followed by irradiation does not decrease the TLK1 activity in HeLa cells. The fact that HeLa model cells lack functional p53 therefore suggests that the decrease in TLK1 activity post IR may be p53 independent [3]. Initially, TLK1 was peptide mapped to be phosphorylated by Chk1 at Ser 695, which lies within the kinase domain [3], but later studies aligned Ser 743 as the site of phosphorylation that marks TLK1 inactivity [14,15]. The sub-cellular localization of full-length human TLK1 in HeLa cells is mostly nuclear with a diffused signal pattern [1]. Post DSB induction with mitomycin C, distinct TLK1 foci that partly co-localized with H2A.X foci 2 h after DNA damage were observed [4]. This suggests that TLK1 plays an important role in DSB repair where most probably it promotes convergence of a hub of factors that function downstream in repair pathways. RAD54 and RAD51 co-localize to form foci with slightly delayed kinetics compared to those formed by TLK1 (approx. 4 h post induction), and RAD54-51 foci persist for 10 h. An intriguing speculation is prompted by the pattern of activity of TLKs upon DNA damage, including IR, and the resulting effect on the phosphorylation of RAD54. As previously mentioned, TLK activity drops rapidly following IR [16] and recovers, and actually over-phosphorylates some of its substrates [4,8], after about one hour in most mammalian cells [8]. This is expected to result in a pattern of rapid de-phosphorylation followed by hyper-phosphorylation of its substrates, including RAD54. This was, in fact, observed for the specific pT700 site in both HeLa and U2OS cells [4], which displayed an initial loss of phosphorylation signal and then a progressive increase during repair. This was accompanied by an interesting pattern of relocalization of GFP-RAD54 and RAD51 (likely in a complex) from the cytoplasm to the nuclei. GFP-RAD54 shuttled back to the cytoplasm after about 10 h, which coincided with the timing of completion of the repair of most DSBs [4].

It is a tempting speculation to imagine that RAD54 (and perhaps in association with ‘inactive’ RAD51 in terms of DNA repair functions) is retained in a cytoplasmic complex by a ‘shuttling’ chaperone that relies on its interaction impinging on the phosphorylation state of RAD54. Following de-phosphorylation of RAD54 after DNA damage and temporary inactivation of TLK1, this results in the shuttling of the complex to the nuclei, where it is recruited to damage sites (namely DSBs) and performs its complex and rather lengthy process of HRR. In time, the activity of TLK1 recovers and re-phosphorylates RAD54, which may be a complex regulatory process accompanying various steps of HRR (most importantly, perhaps, branch migration) and, finally, its return to the cytoplasm after completion of repair. Suggestive evidence that this, in fact, occurs is provided by the fact that the hyper-phosphorylated form of pRAD54-T700 is all nuclear when analyzed by cellular fractionation [4] without any detectable cytoplasmic signal despite the fact that some (pan) RAD54 is detectable in the cytoplasmic fraction [4]. The most likely evidence of the involvement of TLK1 in these dynamic processes, apart from the highly specific role in the phosphorylation of T700, is the fact that if the cells are concomitantly treated with J54 (a highly specific TLK inhibitor [17]) during recovery from IR damage, the GFP-RAD54 is then retained in the nuclei for much longer than the typical 10 h of recovery and does not relocate to the cytoplasm as seen for (J54) untreated cells [4]. The speculation is that a high level of phosphorylation of pRAD54-T700 must persist during various aspect of the HRR process, perhaps starting from the D-loop formation and subsequently during branch migration with its nucleosome displacement, but must be tuned down perhaps during the RAD51 filament dissolution that must accompany the new synthesis of DNA by the repair polymerase complex using the intact homologous chromatid as a template [18]. Perhaps the regulated phosphorylation and de-phosphorylation cycle of RAD54 by TLK1 act as molecular switches to ensure that the right extent of recombinational repair is achieved. Thus, preventing hyper-recombination, which has its own significant drawbacks when viewed in the context of normal replication forks progression [19] or regression/degradation, has recently been found to depend on a new RAD54 function [19].

Another aspect worthy of some discussion and speculation is the possible nature of the two substrates of TLK1 that are known to be directly involved in aspects of DSB repair: RAD9, a sensor of DNA damage that travels along the chromatin to detect lesions (including DSBs), and RAD54, which is specifically recruited to DSBs to perform HRR. While formation of RAD54 IRIFs is detected after some time following IR [20], RAD9, on the other hand, has been found to form foci rapidly post DNA damage, independent of RAD52 group foci localization, and this has been assigned to the checkpoint-related function of RAD9 [21] and potentially to a preferential involvement in NHEJ [22,23,24,25], where it is believed to work as the ‘clamp’ to ensure processivity of the repair polymerases to fill in potentially incompatible ends to facilitate their ligation, a mechanism directly demonstrated in vitro in [26]. While the subject of repair type choice, which clearly involves the particular phase of the cell cycle, has already been much discussed and extensively reviewed [27], we would like to speculate on a possible role of TLK1 in coordinating the cell cycle checkpoint upon IR damage and the repair process itself, irrespective of whether through HRR or NHEJ.

Concurrent with recombinational repair, the DNA damage checkpoint is activated to slow down DNA replication and arrest cells before cell division (G2 phase) until the DNA lesion has been repaired. A functional checkpoint response, first identified in yeast, requires the ATR-related Mec1 kinase and its binding partner Ddc2 (homolog of human ATR-interacting protein ATRIP), as well as the RAD9 checkpoint protein, the RAD53 (checkpoint kinase 2—CHK2) kinase, the Tel1 (ataxia telangiectasia mutated—ATM) kinase, the MRE11/RAD50/Xrs2 (MRX; MRN in humans) complex, the RAD24 (human RAD17)/Rfc2-5 clamp loader, and the Ddc1/Mec3/RAD17 (human RAD9/HUS1/RAD1) DNA clamp [21,28]. These critical checkpoint functions are largely mediated by ataxia telangiectasia and RAD53-related ATR (or its related ATM) and RAD9 to ensure the coordination of the cell cycle progression with sufficient time for repair and, finally, resumption of the cell cycle after deactivation of the checkpoint. TLK1 was found to be critically involved in this process, albeit in higher eukaryotes, on the one hand, through its already discussed role in phosphorylation of RAD9-S328 during restoration of G2 progression after completion of repair [7] and, on the other hand, by mediating the phosphorylation of NEK1-T141, an activating event, thus enabling this key kinase to perform its critical function as a co-activator of ATR upon various types of DNA damage [11,29]. The conservation of proteins related to DDR and repair between yeast and humans made it so that a large number of studies that are (or were) genetically and biochemically much easier to perform with yeast were later mimicked with mammalian models. Of course, there are no TLKs in either S. cerevisiae or S. pombe, but, for example, RAD9 is highly conserved, and its BRCT domain that interacts with TOPBP1 only when phosphorylated at S387 (human) [30] vs. yeast [31], likely by RAD53, suggests very similar regulation of functions.

With different RAD proteins serving as TLK1 substrates, these observations suggest that TLK1 can interact and phosphorylate different RAD proteins and regulate DSB repair, via either HRR or NHEJ, in a concerted and perhaps sequential fashion.

## 3. Role of TLK1 in Regulating HRR Factors

Our recent study showed that TLK1 can phosphorylate RAD54 and regulate different stages of HRR. TLK1 phosphorylates RAD54 at both the N-terminal and C-terminal domains [4]. While the N-terminal domain has been shown to be a regulatory domain of RAD54, the C-terminal domain is a critical domain for contacting the double-strand DNA template and therefore serves as functional domain of the protein. Phosphorylation at both T41 and T59 of RAD54 can serve as an interacting platform for several known proteins of HRR, for example, RAD51AP1, NUCKS1, and CDK2 [32,33,34]. Interestingly, phosphorylation at the C-terminal domain (T700) can alter the intra-protein ionic environment within the Zn-finger-like motif, thereby causing a major change of interaction with its partner protein RAD51. Depleting TLK1, as shown by our previous study and others, exhibits a delayed S-phase progression phenotype [35,36]. HRR is a major participant in replication fork reversal. As recent studies have shown that RAD54 restrains replication fork progression when cells are stressed [19], the phosphorylation of RAD54 by TLK1 could play a significant role in modulating the dynamics of fork progression vs. regression (or even reversal) as a major player for enacting accuracy and processivity at sites where DNA lesions are detected. An earlier explanation for the slow S-phase progression was a defect in chromatin assembly during the replication process due to the established role of TLKs as regulators of Asf1. However, it was also noted that the phenotypes from depletion of TLKs vs. those observed with depletion of Asf1 are quite different with respect to the state of chromatin compaction and cell cycle progression [36], suggesting that TLKs perform rather more complex functions than simply regulating chromatin assembly. For instance, in our previous study, overexpression of a dominant-negative mutant of NEK1, which cannot be phosphorylated by TLK1, implemented an intra S-phase checkpoint upon oxidative stress and may have caused slow S-phase progression [11].

## 4. Role of TLK1 in Eukaryotic Recombination Repair

TLK1 activity is important for homologous recombination repair. Activation of TLK1 activity using gallic acid has been shown to increase HRR activity in cells [37]. TLK1 depletion by siRNA or shRNA methods across different cell lines has been shown to significantly decrease HRR efficiency in cells [4,35]. In a complementary approach, inhibition of TLK1 with thioridazine led to a much greater accumulation of γH2Ax foci due to unrepaired DSBs [38].

The TLK1 interactome reveals that RIF1 is a possible in vivo target [5,39,40]. RIF1 acts at the decision-making junction of NHEJ vs. HRR, downstream of 53BP1 [41,42]. It is known to turn on NHEJ while inhibiting 5′-end resections in HR. Asf1a/b (NTD, 1-154 a.a) has been found to interact with RIF1 (N-terminus, 967-1350 a.a), independent of its chaperone activity [43]. Although TLK1 interacts with both Asf1a/b and RIF1, it remains to be elucidated as to whether TLK1 can impinge on the NHEJ vs. HRR decision by regulating RIF1 and, further, whether it is dependent on TLK1 kinase activity or its chaperone function. In mammalian cells, RIF1 binds to aberrant telomeres in an ATM-53BP1-dependent manner when telomeres are unprotected and recognized as sites of DNA damage [44]. TLK1 depletion has been shown to increase telomeric sister chromatid exchange, thus indicating a state of hyper-recombination [45]. This implies that TLK1 can regulate telomeric recombination as well and possibly mediate chromosome fusion during aberrant cancer hyper-recombination.

TLK1 has been found to interact with another human paralog of RAD54, i.e., RAD54B, which, in yeast (Rdh54), localizes to DNA damage foci in an RAD52-dependent manner [11,21]. In vitro, TLK1 phosphorylates RAD54B at T73 (unpublished data from our lab). Interestingly, the localization of Rdh54 in kinetochores has been shown to be RAD52 independent. Since TLK1 has been shown to function in chromosome segregation in different organisms [35,46,47], it is speculated that TLK1 can regulate RAD54B functions in mitotic or meiotic events of chromosome dynamics. TLK1 preferentially localizes to the nucleolus even without DNA damage induction in HeLa cells [4]. The nucleolus is the compartment that drives ribosomal biogenesis, and, therefore, it is speculated that TLK1 may play a role in ribosome biogenesis or have a specific role in DSB repair at this site, known to be comprised of highly compacted chromatin and highly repetitive sequences that present additional challenges [48].

## 5. TLKs and DNA Damage and Checkpoint Functions

In addition to their functions in the repair of DSBs, TLKs are clearly involved in the repair process of other types of lesions such as UV-induced thymidine dimers (CPDs) [10] and cisplatin-induced ICLs [49]. While we still lack fundamental knowledge of their possible involvement in direct repair of these lesions, a possible explanation for their roles in the repair process, inferred through the effects observed in knock-down and overexpression studies, is, tentatively, their role in chromatin assembly and checkpoint functions. While several studies have described these activities, perhaps the most authoritative one was an unbiased siRNA-mediated screen study for kinases directly affecting potential regulators of recovery from DNA damage in U2OS cells, which positively identified TLK2 as modulating the DDR and G2 recovery. Specifically, these authors reported that TLK2 regulates the Asf1A histone chaperone in response to DNA damage and that its loss of function/depletion, resulting in subsequent Asf1A loss, produces a recovery defect. Both TLK2 and Asf1A are required to restore histone H3 incorporation into damaged chromatin. Failure to do so affects the expression of pro-mitotic genes and compromises the cellular competence to recover from damage-induced cell cycle arrests. Their results demonstrated that TLK2 promotes Asf1A functions during the DNA damage response in G2 to allow for proper restoration of chromatin structure at the break site and subsequent recovery from the arrest. Our early in vitro experiments support these findings, as we showed that TLK1B also affords protection against UV radiation, likely through its effects on chromatin remodeling during NER. We found that nuclear extracts isolated from TLK1B-containing mouse cells promoted more efficient chromatin assembly than comparable extracts lacking TLK1B. TLK1B-containing extracts are also more efficient in the repair of UV-damaged plasmid DNA assembled into nucleosomes. As already mentioned, one of the two earliest known substrates of TLK1 (or TLK1B) is the histone chaperone Asf1, and immuno-inactivation experiments suggested that TLK1B increases repair of UV and other types of damage (see Figure 2) partly through the action of Asf1 on chromatin assembly/disassembly [10].

Furthermore, in TLK1 overexpression studies in normal mouse mammary cells, we directly observed in vivo that where MM3MG-TLK1B-overexpressing cells efficiently repaired almost all the CPDs in 12 h, control MM3MG cells showed very poor levels of CPD removal when assessed using the Southwestern blotting technique [10]. Another method (with intact cells) was also used to compare the repair abilities of the two cell lines. For this purpose, we assessed the repair of episomal vectors in cells over a time course following exposure to UV. MM3MG cells (that were stably transfected with the empty BK-shuttle vector) and the MM3-TLK1B cells (carrying the TLK1B insert in the same vector) were exposed to 5 J/m^2^ of UV radiation and allowed to recover for 0 to 12 h [10] (in Figure 3 of that article). Episomes were extracted at various time points (0, 2, 4, 8, and 12 h) and were either mock-treated or digested with T4 endonuclease V enzyme, which specifically cuts DNA at CPD sites. Since the episomes were damaged on both strands at multiple sites, digestion with T4 endonuclease resulted in extensive cleavage of the recovered plasmid DNA, whereas, in unirradiated (or fully repaired) cells, T4 endonuclease left the plasmids intact. On comparison of the two cell lines (control and TLK1B overexpressing), it was seen that the majority of low-molecular-weight episomal DNA was recovered into repaired high-molecular-weight DNA in the MM3G-TLK1B cells between 8 and 12 h, while the T4-endonuclease-treated episomal DNA still appeared as a low-molecular-weight smear in the MM3MG cells. These two experiments strongly suggest that the MM3G-TLK1B cells repaired DNA faster than the control MM3MG cells. Perhaps, the explanation for these effects of TLK1B in UV damage repair is much more complex than the simple attribution of its activity in chromatin remodeling and rather implies that there are still direct targets/substrates of TLKs that are involved in the actual repair process, at least in the case of NER [10].

In addition to the aforementioned established role of TLK1 in the repair of cyclo-pyrimidine dimers (CPDs), there is emerging evidence for a role in cisplatin-induced ICLs and base adducts, particularly inferred through the effects observed in knock-down studies of cancer cells. Cholangiocarcinoma is a particularly devastating carcinoma with limited treatment options. As tousled-like kinase 1 (TLK1) is a serine/threonine protein kinase that regulates DNA replication and DNA repair pathways, these authors studied its possible role in sensitization to cisplatin, which is one of the few treatment options for this deadly disease, in addition to gemcitabine [50]. First, these authors showed that TLK1 is abundantly expressed in cholangiocarcinoma as well as in cell lines derived from cholangiocarcinoma. Second, although siRNA knockdown of TLK1 did not affect the viability of cholangiocarcinoma cells, it sensitized these to cisplatin-induced apoptosis. Furthermore, immunofluorescence analysis of γH2AX foci showed that silencing of TLK1 enhanced DNA damage induced by cisplatin treatment. Their results clearly support the hypothesis that TLK1 plays a pivotal role in the repair of cisplatin-induced DNA damage.

## 6. Role of TLK1 in Cancer

In different disease models such as prostate cancer (PCa) and glioblastoma (GBM) models, TLK1 depletion has been shown to elicit DNA damage response, and, therefore TLK1 forms a druggable target [51,52,53] (Figure 2). Human cancer data revealed frequent up-regulation of TLK genes and an association with poor patient outcomes in multiple types of cancer, and depletion of TLK activity led to increased replication stress and DNA damage in a panel of cancer cells [36]. GWAS analyses revealed that TLK1 mutations are rare in cancer, but its overexpression is frequently linked to poor prognosis [45,54]; this is particularly evident for patients with low Gleason scores (e.g., GS = 6—Ualcan.path.uab.edu/analysis page 2) who would otherwise be expected to fare better in terms of survival. Likewise, amplification of TLK2 has been reported as a frequent event in breast cancer [55], GBM [56], and neuroblastoma [57]. Our previous work showed that DNA damage activates the TLK1 > NEK1 > YAP1 axis, which either further elevates the apoptotic pathway in PCa [20] or can lead to compensatory adaptation to genotoxins [17]. Broadly, it can be said that TLKs can impact cancer ontology and/or progression by regulating genome and epigenome stability [5], as well as potentially suppressing aspects of innate immune signaling [45]. Specifically, on this important interplay of the critical relation between cancer progression and the immune response, it was reported that depletion of TLK activity, apart from causing replication stress, increases chromosomal instability and cell cycle arrest or death. These studies also showed that stalled forks in TLK-depleted cells are processed by BLM, SAMHD1, and the MRE11 nuclease to generate ssDNA-activating checkpoint signaling. Thus, TLK depletion ultimately impairs heterochromatin maintenance, inducing features of alternative lengthening of telomeres and increasing spurious expression of other repetitive elements associated with impaired deposition of the histone variant H3.3 [5]. Moreover, TLK2 depletion culminates in a BLM-dependent, STING-mediated innate immune response, whereby, in many human cancers, TLK1/2 expression correlates with signatures of chromosomal instability and anti-correlates with STING and innate and adaptive immune response signatures [45]. All this prompts an exciting but challenging new prospective view of how TLKs may operate in terms of different, albeit still understudied, aspects of their involvement in cancer development and progression and as a possible target in cancer immunotherapy and maybe in immune checkpoint regulation [58,59], recombination, and cytokine signaling [5]. Further important genetic evidence came from the identification of a novel heterozygous variant of TLK1 in a patient with a neuro-developmental and immunological syndrome. This mutation impaired kinase activity and caused alterations in genes involved in class switch recombination, suggesting its potential involvement in central nervous system and immune system development (Travis Stracker, personal communication).

## 7. Conclusions: Targeting TLK1 for Cancer Treatment

After the seminal discovery that TLK1 has an important modulatory role in DDR and DNA repair, the quest turned rapidly into trying to identify specific TLK inhibitors to enhance the effectiveness of XRT or radiomimetic drugs. This led to the initial identification of certain phenothiazine (PTH) antipsychotics as surprisingly specific TLK inhibitors that, in fact, enhanced the killing of cancer cells, in vitro and in xenografts, when combined with IR or doxorubicin [38]. A newer PTH scaffold, called J54, that substantially lacks any anti-dopaminergic undesirable effect, showed very promising results in the regression of androgen-sensitive prostate cancer cells largely by passing the DDR and thereby enforcing entry into catastrophic mitotic progression, even resulting in substantial tumor regression in SCID xenografts [17]. Other studies on the structure/function of TLKs have revealed additional potential inhibitors [60]. Considering the availability of such drugs, of even greater importance now is the rebound effort to target the activity of TLK1 in RAD9 and RAD54, with the obvious suggestion that translesion (TSL) repair, single-strand gaps, and HRR can be simultaneously targeted. Various chemotherapeutic agents, therefore, fall under the umbrella of TLK inhibitors for their therapeutic potentiation. These include, for example: topoisomerase poisons, bleomycin, MMC, PARPis, and cisplatin, all of which ultimately lead to the formation of SSBs and DSBs. For some of these (e.g., doxorubicin and cisplatin), direct evidence of synthetic lethality in combination with TLK inhibitors has already been verified [38,61]. Of particular note is the reported synthetic lethal interaction of combining the knock down of TLKs with PARP inhibitors [36], which could be expected to engender a double hit on both NHEJ and HRR based on the already described functions of TLKs. Inhibiting TLK1 prior to radiation in diseases such as GBM, where base excision repair is activated [53], or cholangiocarcinoma or prostate cancer, where intra-strand crosslink repair is elevated [49], increases their chemosensitization, which suggests that TLK1 forms a nexus of a vast spectrum of DNA repair mechanisms (Figure 2). In addition to these obvious actionable targets via inhibitors of TLK1 in DNA repair and checkpoint surveillance, more recent functions for TLK1 in disparate, important oncogenic pathways, such as regulation of AKT (via AKTIP), the important pro-metastatic kinase MK5, and as the ultimate effector of the Hippo pathway, YAP (via NEK1), are coming to light (reviewed in [62]). In addition to these, a glimpse into a potentially important role of the TLK1 > NEK1 interaction in an anti-apoptotic function related to VDAC1, a key regulator of mitochondrial permeability, was recently provided in our studies [63], wherein we speculated that a limiting shuttling function of NEK1 between nuclei and mitochondria to regulate competing activities in either compartment could be a critical decision point between implementing DNA repair vs. prompting apoptosis if the damage is too extensive [63]. In all, the important roles of TLKs in various aspects of oncogenic development and as possible targets for various cancer-directed therapies are slowly but surely becoming ‘untousled’.

## Figures and Tables

**Figure 1 ijms-24-13369-f001:**
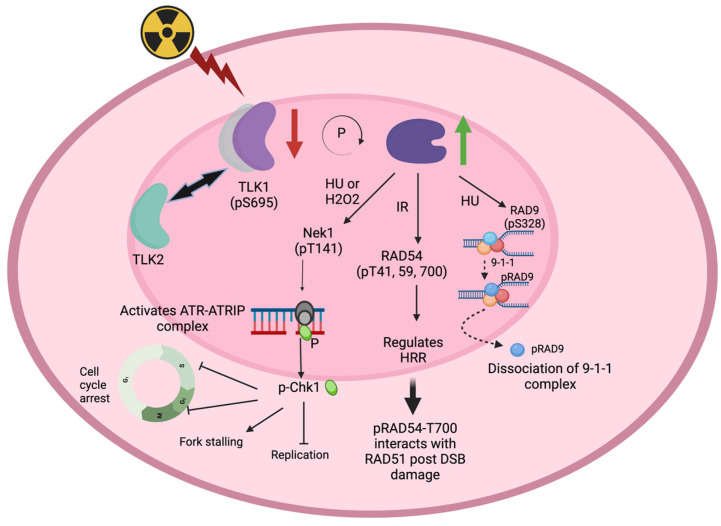
Mechanistic role of TLK1 in DNA damage repair (DDR). TLK1 (light purple) can heterodimerize with TLK2 (cyan) in certain cell types, and normal kinase activity of TLK1 is dependent on homodimerization/oligomerization and/or heterodimerization. The N-terminal region of TLK1/TLK2 is essential for higher order arrangement. In cells, when DSB is induced with IR, there is a transient decrease in TLK1 activity (red arrow) which is restored within an hour (in HeLa cells). The inhibition of TLK1 is dependent on ATM-Chk2 phosphorylation at S695 (or equivalent S457 of TLK1B, spliced isoform). Activation of TLK1 (dark purple) is dependent on its autophosphorylation (shown by circular arrow). When TLK1 is active following DSB damage (green arrow), TLK1 phosphorylates multiple DDR substrates. TLK1 can phosphorylate RAD54, particularly at T41, T59, and T700, which has been investigated recently [4]. During replication stress by hydroxyurea (HU) or oxidative damage by H2O2, TLK1 can phosphorylate Nek1 at T141, which leads to ATR–ATRIP complex (gray spheres) activation, which functions upstream of Chk1 (green sphere). Chk1 is phosphorylated within the nucleus and inhibits replication, leads to fork stalling, and arrests the cell cycle at G1/S, intra-S phase, and G2/M checkpoints. During HU-dependent stress, RAD9 (light blue) is phosphorylated by TLK1 at S328 in the 9–1–1 complex at the replication fork. Phosphorylated RAD9 (dark blue) dissociates from the 9–1–1 complex and shuttles to the cytoplasm after recovery of DNA damage. Figures were designed in BioRender.com (accessed on 26 July 2023).

**Figure 2 ijms-24-13369-f002:**
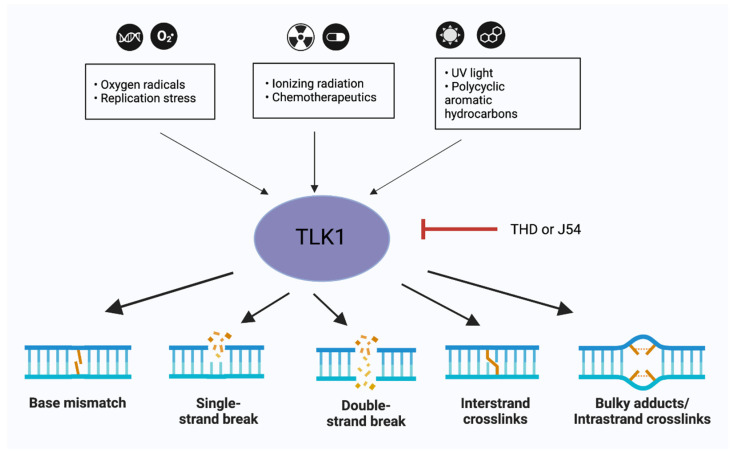
TLK1 at the core of DNA repair mechanisms. TLK1 forms a therapeutic target in multiple tissue-specific cancer models. Inhibiting TLK1 can affect the base excision repair, single-strand break (SSB) repair, double-strand break (DSB) repair, or inter/intra-strand crosslink (ICL) repair in cells. The small black arrows indicate DNA-damaging agents which activate TLK1, while the bold black arrows indicate that inhibiting TLK1 affects the specific DNA repair pathways. Figures were designed in BioRender.com.

## Data Availability

No new unpublished data were included in this review.

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
