# Peer review of "Untousling the Role of Tousled-like Kinase 1 in DNA Damage Repair"

_ijms, 2023, doi:10.3390/ijms241713369_

Round 1
Reviewer 1 Report
The review covers a large body of work and is comprehensive. A few minor comments about specific statements to consider.
1. The authors stated, “..with mouse embryos, which showed that TLK1 is important for later development stages”. I do not recall any data showing defects in later developmental stages in these mice. KOMP data also does not indicate any issues. Please clarify.
2. The authors stated “One such substrate is Asf1b which is phosphorylated by TLK1/B”- remove the backslash so it is TLK1B.
3. Add comma after “in vitro” in the line “In vitro Tousled like Kinase 1 can bind directly with…”.
4. The authors state “TLK1 can phosphorylate RAD54 particularly at T41, T59 and T700 which has been investigated in the current study.” It was not investigated in this review paper- should reference where it was investigated.
5. The authors state ”Upon DSB induction by IR, TLK1 is transiently inhibited via a Chk1 dependent mechanism, but it is important to note that depletion of Chk1 followed by irradiation did not decrease the TLK1 activity in HeLa cells. Further, HeLa model cells lack functional p53 and therefore, the decrease in TLK1 activity post IR may be p53 independent (3).” I am confused by the point here- saying it is important to note would suggest the data is in opposition, when it in fact supports a role for CHK1 in inhibiting TLK1. Regarding the last line, has someone suggested this is p53 dependent? Unclear on what the relevance of this is since p53 does not affect CHK1 activation.
6. The authors state “As previously mentioned, TLK activity drops rapidly following IR (15) and only recovers, actually overshoots, after about one hour in most mammalian cells (16).” Unclear what is meant by “actually overshoots”.
7. The authors state “This is accompanied by the peculiar pattern of relocalization of GFP-RAD54 and RAD51 (likely in a complex) from the cytoplasm to the nuclei, …”. Do they mean peticular? Unclear why it is peculiar.
8. Change detect to detected in the following line “While formation of RAD54 IRIFs is detect after some time following IR (21),”
9. Unclear why the authors decided to use yeast nomenclature in the following section, seems unnecessary- “A functional checkpoint response requires the ATR-related Mec1 ki-nase and its binding partner Ddc2 (homolog of human ATRIP), as well as the Rad9 check-point protein, the Rad53 (CHK2) kinase, the Tel1 (ATM) kinase, the Mre11/Rad50/Xrs2 (MRX; MRN in humans) complex, the Rad24 (human RAD17)/Rfc2-5 clamp loader, and the Ddc1/Mec3/ Rad17 (human RAD9/HUS1/RAD1) DNA clamp ((22,28)).”
10. The authors state “An earlier explanation for the slow S-phase progression was attributed to a defect in chromatin assembly during the replication process due to the established role of TLKs as regulators of Asf1. However, it was also noted that the phenotypes from depletion of TLKs vs those observed with depletion of Asf1 are quite different with respect to the state of chromatin assembly and cell cycle progression (36).” As I understand it, the main difference in phenotype is the lack of ssDNA generation in ASF1 depletion that was previously attributed to potential differences in MCM progression by Groth -MCM is proposed to drive uncoupling that generates ssDNA- but both ASF1 and TLK depletion cause defects in fork progression. Moreover, it is clear that other things can phosphorylate ASF1 with data from Tyler implicating DNA-PKcs and data from Dutta implicating CHK1, also potentially explaining the more robust effect of ASF1 vs TLK depletion.
11. Unclear what is going on with this sentence: Ualcan.path.uab.edu/analy-sis page 2), which would otherwise be expected to fare better survival.
12. The authors state “This led to the initial identification of certain phenothiazines (PTH) antipsychotics as surprisingly specific TLK inhibitors that in fact enhanced the killing of cancer cells, in vitro and xenografts, when combined to IR or doxorubicin (39).” Phenothiazines are reported to inhibit a large number of things. Recent work from Lee et al did not observe inhibitory effects of THD (PMID: 34662748). How clear is the specificity data?
Aside from minor issues, the English is clear. Pointed out a few things in the review. Journal editors should handle this, not me.
Author Response
Reviewer 1: We sincerely thank Rev.1 for the excellent suggestions. Detailed answers are listed below:
- The authors stated, “..with mouse embryos, which showed that TLK1 is important for later development stages”. I do not recall any data showing defects in later developmental stages in these mice. KOMP data also does not indicate any issues. Please clarify.
We would like to correct the statement as follows- The overall function of TLK1 and TLK2 in previously cited article was addressed with viability studies with mouse embryos, which showed that TLK1 is not essential during the placental development stages whereas TLK2 is required for placental development. Rather, TLK1 biological role was implicated for sensitivity to DNA damage induction, more so in the conditional KO of both proteins.
- The authors stated “One such substrate is Asf1b which is phosphorylated by TLK1/B”- remove the backslash so it is TLK1B.
We corrected as per the suggestion.
- Add comma after “in vitro” in the line “In vitro Tousled like Kinase 1 can bind directly with…”.
We corrected as per the suggestion.
- The authors state “TLK1 can phosphorylate RAD54 particularly at T41, T59 and T700 which has been investigated in the current study.” It was not investigated in this review paper- should reference where it was investigated.
We added the reference in Figure 1 legend- TLK1 can phosphorylate RAD54 particularly at T41, T59 and T700 which has been investigated recently (Ghosh et al., 2023).
- The authors state ”Upon DSB induction by IR, TLK1 is transiently inhibited via a Chk1 dependent mechanism, but it is important to note that depletion of Chk1 followed by irradiation did not decrease the TLK1 activity in HeLa cells. Further, HeLa model cells lack functional p53 and therefore, the decrease in TLK1 activity post IR may be p53 independent (3).” I am confused by the point here- saying it is important to note would suggest the data is in opposition, when it in fact supports a role for CHK1 in inhibiting TLK1. Regarding the last line, has someone suggested this is p53 dependent? Unclear on what the relevance of this is since p53 does not affect CHK1 activation.
We edited the statement as follows- Upon DSB induction by IR, TLK1 is transiently inhibited via a Chk1 dependent mechanism, and hence depletion of Chk1 followed by irradiation did not decrease the TLK1 activity in HeLa cells. The fact that HeLa model cells lack functional p53 therefore suggests that, decrease in TLK1 activity post IR may be p53 independent.
- The authors state “As previously mentioned, TLK activity drops rapidly following IR (15) and only recovers, actually overshoots, after about one hour in most mammalian cells (16).” Unclear what is meant by “actually overshoots”.
We corrected and removed the phrase to over phosphorylates some of its substrates.
- The authors state “This is accompanied by the peculiar pattern of relocalization of GFP-RAD54 and RAD51 (likely in a complex) from the cytoplasm to the nuclei, …”. Do they mean peticular? Unclear why it is peculiar.
We found that RAD54-GFP in HeLa is cytoplasmic and localizes to nucleus post DNA damage which was an interesting observation and therefore we corrected the statement as follows- This is accompanied by the interesting pattern of relocalization of GFP-RAD54 and RAD51 (likely in a complex) from the cytoplasm to the nuclei, and then back to the cytoplasm after about 10 hours, which coincides with the timing of completion of the repair of most DSBs.
- Change detect to detected in the following line “While formation of RAD54 IRIFs is detect after some time following IR (21),”
We corrected the grammar- While formation of RAD54 IRIFs is detected after some time following IR.
- Unclear why the authors decided to use yeast nomenclature in the following section, seems unnecessary- “A functional checkpoint response requires the ATR-related Mec1 ki-nase and its binding partner Ddc2 (homolog of human ATRIP), as well as the Rad9 check-point protein, the Rad53 (CHK2) kinase, the Tel1 (ATM) kinase, the Mre11/Rad50/Xrs2 (MRX; MRN in humans) complex, the Rad24 (human RAD17)/Rfc2-5 clamp loader, and the Ddc1/Mec3/ Rad17 (human RAD9/HUS1/RAD1) DNA clamp ((22,28)).”
The conservation of proteins related to the DDR and Repair between yeast and human made it so that a large number of studies that are (or were) genetically and biochemically much easier to do with yeast, were later mimicked with mammalian models. Of course, there are no TLKs in either S.cerevisiae or S. pombe, but for example Rad9 is highly conserved and its BRCT domain that interacts with TOPBP1 only when phosphorylated at S387 (human -https://elifesciences.org/articles/39979) or S129 (yeast - https://www.ncbi.nlm.nih.gov/pmc/articles/PMC1973948/) , likely by Rad53, suggests very similar regulation of functions.
- The authors state “An earlier explanation for the slow S-phase progression was attributed to a defect in chromatin assembly during the replication process due to the established role of TLKs as regulators of Asf1. However, it was also noted that the phenotypes from depletion of TLKs vs those observed with depletion of Asf1 are quite different with respect to the state of chromatin assembly and cell cycle progression (36).” As I understand it, the main difference in phenotype is the lack of ssDNA generation in ASF1 depletion that was previously attributed to potential differences in MCM progression by Groth -MCM is proposed to drive uncoupling that generates ssDNA- but both ASF1 and TLK depletion cause defects in fork progression. Moreover, it is clear that other things can phosphorylate ASF1 with data from Tyler implicating DNA-PKcs and data from Dutta implicating CHK1, also potentially explaining the more robust effect of ASF1 vs TLK depletion.
We agree with reviewers referenced citations and would like to add as follows in the manuscript text-
However, it was also noted that the phenotypes from depletion of TLKs vs those observed with depletion of Asf1 are quite different with respect to the state of chromatin compaction and cell cycle progression (39), suggesting that TLKs rather perform more complex functions than simply regulating chromatin assembly. We would rather like to mention that from our previous study lack of TLK1>NEK1>ATR regulation implements an intra S-phase checkpoint and may cause the slow S-phase progression(11).
- Unclear what is going on with this sentence: Ualcan.path.uab.edu/analy-sis page 2), which would otherwise be expected to fare better survival.
It is the second page of the Ualcan analysis of PCa Kaplan Meyer curve, stratified with respect to GS (link below). The patients with low GS are expected to do better than those with high GS (more advanced disease cellular features). But those that have high TLK1 expression do terrible despite the pathology "good prognosis" branding.
https://ualcan.path.uab.edu/cgi-bin/TCGA-survival1.pl?genenam=TLK1&ctype=PRAD
- The authors state “This led to the initial identification of certain phenothiazines (PTH) antipsychotics as surprisingly specific TLK inhibitors that in fact enhanced the killing of cancer cells, in vitro and xenografts, when combined to IR or doxorubicin (39).” Phenothiazines are reported to inhibit a large number of things. Recent work from Lee et al did not observe inhibitory effects of THD (PMID: 34662748). How clear is the specificity data?
This is very complex issue to address, particularly since it deviates from the main topics we are discussing in this review while dealing with the TLKs activity on the DDR. But I will provide some partial explanations below:
- Concerning the inhibition (or rather lack of) of MDA-MB231 cells by the Lee paper, I really cannot comment on this, since it is totally contractor to the majority of earlier studies, starting perhaps with this https://www.researchgate.net/publication/21751499_Tamoxifen-resistant_human_breast_cancer_cell_growth_Inhibition_by_thioridazine_pimozide_and_the_calmodulin_antagonist_W-13/link/0046353741422e53b0000000/download
- Concerning the apparent lack of inhibition of phosphorylation of Asf1 by THD in IVK assays, this is a very complex issue. Besides the fact that we do not think that Asf1 is the best substrate for TLKs in vitro (which is why we soon shifted to the RAD9-S328 peptide or NEK1-T141), we recently reported that the inhibition of TLK1 vs. TLK1-kinase domain is very complex in vitro vs in in vivo (in cells). Where a lot has to do with the autoactivation process. For instance, the TLK1-KD (only), is about 1000 times more active (Kcat) than the full length protein, at least in the initial period of the ATPase reaction, and is poorly inhibited by all the PTH derivatives we have tested. But in vivo, using pNEK1 as relay, several PTHs were inhibitory in the 200 nM range: https://pubs.rsc.org/en/content/articlelanding/2023/ob/d2ob02191a
- The broad effects reported for phenothiazines (PTH) need to be revisited. For example, some of their growth-inhibitory effects were specifically attributed to their action on calmodulin (https://pubmed.ncbi.nlm.nih.gov/3457971/) and particularly in circumventing drug resistance (like to doxorubicin). However, the activity of PTH in cells (in vivo) is seen in the low uM range, whereas the inhibition (ED50) of calmodulin in vitro is in the 50-100 uM. Difficult to explain since these drugs readily pass through the cell membrane.
Reviewer 2 Report
The Review article by Ghosh and De Benedetti focuses on the role of tousled like kinase 1 in DNA repair. Several DNA repair pathways and multiple other DNA repair factors are mentioned, however, sometimes the structure and depth of the text could be improved.
Specific points.
1. In the Abstract, some abbreviations are not spelled, e.g. DDR, DSB, TLK. One may guess the meaning from the context or if have a background in the DNA repair field. Usually, the abbreviations should be introduced in 1) Abstract, 2) Main text, and 3) Figure legends.
2. If the authors use italics in the text (e.g. for species), it would make sense to also use italics for the latin expressions, e.g. in vivo, in vitro.
3. Figure 1 has the potential to be centered and larger.
4. Kindly spell abbreviations in the text when used the first time. For example, IR, ATM, Chk2, HU, ATR, ATRIP, Chk1, Rad1/9/17/24/50/51/52/53/54, GFP, HRR, NEK1, AKTIP, FANCM, IRIF, Mec1/3, Ddc2, Tel1, Mre11, Xrs2, MRN, Rfc2-5, HUS1, AP1, NUCKS1, CDK2, Asf1, siRNA, shRNA, RIF1, 53BP1, NTD, a.a., Rdh54, CPD, ISL, NER, MM3MG, GWAS, YAP1, BLM, SAMHD1, STING, SCID, MMC, PARP, etc.
5. Very long paragraphs are hard to read. A general expectation is to have the paragraphs shortened or divided.
6. End of page 3, "that are known to be directly known to be involved is aspects". Kindly check.
7. Page 4. Reference: Maranon 2020 #221. Kindly check.
8. Page 5 refers to "unpublished data". Kindly check if the journal allows unpunished data, especially for literature reviews. In any case, it would be necessary to mention to whom these unpublished data belong.
9. Kindly check if the spelling is consistent, e.g. Mre11 vs MRE11.
10. "Conclusion" section is too long. Conclusions are expected to be sharper and more concise.
11. Kindly check if references 17 and 32 are the same.
12. Reference 36 does not show the date (2018?). Also, reference 38.
13. Overall, there are too many typos, and the manuscript appears as an initial draft rather than a solid final paper.
There are dozens of typos and inconsistencies. Most of the abbreviations are not spelled. If used only once or twice, some abbreviations are not needed and can be replaced with full-size words.
Author Response
Reviewer 2: Thanks for the excellent suggestions. Detailed responses below:
Specific points.
- In the Abstract, some abbreviations are not spelled, e.g. DDR, DSB, TLK. One may guess the meaning from the context or if have a background in the DNA repair field. Usually, the abbreviations should be introduced in 1) Abstract, 2) Main text, and 3) Figure legends.
Thank you for the suggestion. We agree and we provided a list of abbreviations below the Abstract section. Also, we mentioned full-form at first time mention in Main text section and Figure legends.
- If the authors use italics in the text (e.g. for species), it would make sense to also use italics for the latin expressions, e.g. in vivo, in vitro.
We changed the font to italics for ‘in vitro’ and ‘in vivo’ terms.
- Figure 1 has the potential to be centered and larger.
We changed the alignment of Figure 1 and centered the Figure 1.
- Kindly spell abbreviations in the text when used the first time. For example, IR, ATM, Chk2, HU, ATR, ATRIP, Chk1, Rad1/9/17/24/50/51/52/53/54, GFP, HRR, NEK1, AKTIP, FANCM, IRIF, Mec1/3, Ddc2, Tel1, Mre11, Xrs2, MRN, Rfc2-5, HUS1, AP1, NUCKS1, CDK2, Asf1, siRNA, shRNA, RIF1, 53BP1, NTD, a.a., Rdh54, CPD, ISL, NER, MM3MG, GWAS, YAP1, BLM, SAMHD1, STING, SCID, MMC, PARP, etc.
We have cited the references for the terms of the proteins where it originated. However, it is not possible to spell the abbreviations for all the proteins (for e.g- Rad1/9/17/24/50/51/52/53/54, Mec1, Ddc2)
- Very long paragraphs are hard to read. A general expectation is to have the paragraphs shortened or divided.
Thank you for the suggestion. We made the necessary changes of shortening paragraphs and sentences.
- 6. End of page 3, "that are known to be directly known to be involved is aspects". Kindly check.
We corrected the sentence as follows- that are known to be directly involved in aspects of DSB repair
- Page 4. Reference: Maranon 2020 #221. Kindly check.
We corrected with right citation format.
- Page 5 refers to "unpublished data". Kindly check if the journal allows unpunished data, especially for literature reviews. In any case, it would be necessary to mention to whom these unpublished data belong.
We understand and corrected as follows- unpublished data from our lab
- Kindly check if the spelling is consistent, e.g. Mre11 vs MRE11.
We edited to MRE11.
- "Conclusion" section is too long. Conclusions are expected to be sharper and more concise.
The fact that TLK1 from past studies have manifested a wide role in different axes of DNA damage repair, contrasted with its apparent lack of overt phenotypes in KO-mice, required to be briefed in the Conclusion section.
- Kindly check if references 17 and 32 are the same.
We beg to differ that the references are different.
- Reference 36 does not show the date (2018?). Also, reference 38.
- Overall, there are too many typos, and the manuscript appears as an initial draft rather than a solid final paper.
Round 2
Reviewer 2 Report
The authors have revised the Review article according to the Reviewers' suggestions. Not all recommendations were followed or commented on, however, some explanation is provided to justify the authors' point of view.
There is a potential to proofread the text. Usually, the journal does it, however, it will be even better if authors catch most of the typos and format differences to ensure the best possible final version of the manuscript.
Several typos need to be corrected